# Effects of Traditional Chinese Exercise on Oxidative Stress in Middle-Aged and Older Adults: A Network Meta-Analysis

**DOI:** 10.3390/ijerph19148276

**Published:** 2022-07-06

**Authors:** Delong Chen, Guanggao Zhao, Jingmei Fu, Shunli Sun, Xiaoxiao Huang, Liqiang Su, Zihao He, Ting Huang, Ruiming Chen, Xuewen Hu, Tianle Jiang, Minghui Quan

**Affiliations:** 1School of Physical Education, Nanchang University, Nanchang 330031, China; delongchen1996@outlook.com (D.C.); xiaoxiaohuang1030@outlook.com (X.H.); salmanliu6@163.com (T.H.); ruimingchen1021@outlook.com (R.C.); ndhuxuewen@outlook.com (X.H.); tianle1127@outlook.com (T.J.); 2Jiangxi Sports Science Medicine Center, Nanchang 330006, China; fujinmei2022@outlook.com (J.F.); sunsl087@outlook.com (S.S.); 3Physical Education College, Jiangxi Normal University, Nanchang 330022, China; s-2005100153@163.com; 4School of Sports and Human Sciences, Beijing Sport University, Beijing 100091, China; hudson20142@hotmail.com; 5School of Kinesiology, Shanghai University of Sport, Shanghai 200438, China

**Keywords:** traditional Chinese exercises, oxidative stress, network meta-analysis

## Abstract

Objective: To evaluate the best option among traditional Chinese exercises for reducing oxidative stress in middle-aged and older adults, using a network meta-analysis. Methods: PubMed, Web of Science, and CNKI databases were used. We searched randomized controlled trials (RCTs) on middle-aged and older adults to influence oxidative stress by any traditional Chinese exercises from the beginning to 20 January 2022. A network meta-analysis of randomized control trials was performed comparing the changes in the concentration of glutathione peroxidase (GPX), malondialdehyde (MDA), and superoxide dismutase (SOD) as primary outcomes, following different therapeutic interventions with traditional Chinese exercises in middle-aged and older adults over 30 years old. Standardized mean differences (SMD) and 95% confidence intervals (CI) were used to assess the correlation between each group of interventions, and surface under the cumulative ranking (SUCRA) was used to rank the best interventions. Results: The meta-analysis comprised 15 trials with a total of 927 participants and six interventions: (Wuqinxi (WQX), Baduanjin (BDJ), Tai Ji Quan (TJQ), Yijinjing (YJJ), Mawangdui Daoyin (MWD), and no exercise intervention (NEI)). Regarding GPX: WQX [SMD = 2.79 (1.75, 3.83)], TJQ [SMD = 0.47 (0.23, 0.70)], YJJ [SMD = 1.78 (1.18, 2.37)], MWD [SMD = 1.89 (1.36, 2.43)] were superior in increasing GPX relative to NEI. Regarding MDA: WQX [SMD = 1.68 (0.94, 2.42)], YJJ [SMD = 0.99 (0.28, 1.69)] were superior in reducing MDA relative to NEI. Regarding SOD: WQX [SMD = 1.05 (0.10, 2.01)] were superior in increasing SOD relative to NEI. WQX topped the SUCRA with GPX: 0.97, MDA: 0.91, and SOD: 0.94. Furthermore, WQX was more effective than TJQ in interfering with GPX [SMD = 2.32 (1.26, 3.39)] and MDA [SMD = 1.47 (0.26, 2.67)], and a significantly better intervention effect on SOD than YJJ [SMD = 1.52 (0.80, 2.24)] and MWD [SMD = 0.89 (0.03, 1.75)]. Conclusion: Traditional Chinese exercise can help middle-aged and older adults reduce oxidative stress. WQX may be the best traditional Chinese exercise of the exercises evaluated for reducing oxidative stress in middle-aged and older adults.

## 1. Introduction

There is evidence that oxidative stress is linked to cancer [1], cardiovascular disease [2], diabetes [3], kidney disease [4], neurodegeneration [5], and other non-communicable diseases. On the other hand, oxidative stress is more likely to occur as people get older [6]. According to data, the world’s elderly population will number 722 million in 2020, up by 24 million from 2019 [7]. Faced with population aging and the resulting rise in non-communicable diseases caused by oxidative stress, many scholars have devoted themselves to antioxidative stress intervention research, including antioxidants and exercise. Large doses of antioxidant supplements pose a particular risk among these, and the impact of antioxidants on the human body is currently unknown [8]. On the other hand, exercise is an effective non-pharmacological treatment option in a global context. Numerous studies have demonstrated that sustained exercise can increase the body’s antioxidant capacity while simultaneously reducing the level of oxidative stress [9]. From the long-term perspective, maintaining a particular level of physical activity habits may prevent oxidative stress from progressing to an undesirable state, lowering the risk of cardiovascular disease, diabetes, and other oxidative stress-related non-communicable diseases.

It is important to note that traditional Chinese exercise seems to be more effective than other exercise modalities at reducing oxidative stress indicators [10]. The study found that whether it is Wuqinxi (WQX) [11], Baduanjin (BDJ) [12], Tai Ji Quan (TJQ) [13], Yijinjing (YJJ) [14], etc., it has significant effects on oxidative stress. However, each of these traditional Chinese exercises has its unique style, and the impact of interventions on different oxidative stress indicators may differ. For example, the WQX mimics the movements performed by the five animal behaviors, exercise emphasizes breathing control and mental regulation during exercise [11]. The BDJ is a full stretching exercise through the upper and lower trunk [12]. The TJQ emphasizes the stability of lower limb movements [13]. The YJJ emphasizes the integration of consciousness and body [14]. Their differences are of great value for the formulation of personalized exercise prescriptions.

The primary cause of the synthesis of oxidative stress is a decrease in cytochrome c oxidase activity brought on by mitochondrial dysfunction, which results in peroxidation damage to the mitochondrial membrane and the production of hyperactive species [15]. When active species are produced in excess, cell structure, lipids, proteins, and genetic material are all destroyed [16]. At the same time, middle-aged and older adults will be at higher risk of oxidative stress as they age [17]. Furthermore, with the enormous increase in the world’s middle-aged and older population, the level of oxidative stress in middle-aged and older adults has received much attention from international scholars.

There isn’t currently a best practice for how different traditional Chinese exercise workouts affect middle-aged and older adults’ oxidative stress levels. Therefore, a network meta-analysis will be employed in this study to examine the effects of several traditional Chinese exercise regimens on oxidative stress levels in middle-aged and older adults as a foundation for future targeted health promotion interventions.

## 2. Methods

### 2.1. Registration

Our research program has been registered on Prospero, the International Register of Expectations for System Evaluation; Registration number: CRD42022332724.

### 2.2. Search Strategy

This study used the summary stowage of traditional Chinese exercises by the Chinese National Sports Administration as the basis for the retrieval of interventions. Total antioxidant capacity (TAC), glutathione peroxidase (GPX), malondialdehyde (MDA), and superoxide dismutase (SOD) indexes were used to measure oxidative stress [18].

According to PRISMA principles [19], electronic databases, including PubMed, Web of Science, and the China National Knowledge Infrastructure (CNKI), were systematically searched from inception to 20 January 2022. The search referred to the random controlled trial (RCT) search strategy described in the Cochrane Systematic Evaluation Manual (version 5.1.0, Cochrane, London, UK, https://training.cochrane.org/, accessed on 27 June 2022), and used a combination of subject terms and free words to search for each synonym and adjusted according to the specific database. The search terms consisted of: (1) Participants: Age *, old *, elder *. (2) Intervention: Wuqinxi, Wu qin xi, five animal exercises, five-animal play, Baduanjin, Ba duan jin, eight pieces of brocade, eight silken movements, eight section brocade, Yijinjing, tendon change classic, Taichi, Tai chi, Tai ji, T’ai chi, Taijiquan, Taichiquan, Liuzijue, Sixzijue, Mulanquan, Mawangdui, Mawang, Qigong, Qi gong, Chi kung, Qi gung, traditional Chinese exercise, CTE. (3) Outcome: Total antioxidant capacity, TAC, T-AOC, glutathione peroxidase, GPX, GPH-X, malondialdehyde, MDA, superoxide dismutase, SOD. (4) Study design: RCT, trial.

### 2.3. Inclusion and Exclusion Criteria

According to the population, intervention, comparison, outcomes, and study designs (PICOS) structure [20], the inclusion criteria were: (1) Participants: Age ≥ 30, middle-aged, older. (2) Interventions: Interventions using traditional Chinese exercises (WQX, BDJ, TJQ, YJJ, etc.) in the pilot group, and the length of the intervention was a long-period intervention of >4 weeks. (3) Control: There was at least one control group with the control intervention of no exercise intervention (NEI). (4) Outcome: The biomarkers widely used to assess OS level, TAC, GPX, and MDA, as well as a biomarker of antioxidant capacity, SOD, are employed as outcome indicators in this research. (5) Study Design: RCT.

Exclusion criteria: (1) Literature in non-Chinese or non-English; (2) review literature, literature with missing critical data information; (3) use of combined therapeutic interventions (e.g., TJQ health gong combined with negative air oxygen ion inhalation); (4) studies using the same data.

### 2.4. Data Extraction

Two researchers (Chen and Chen) independently screened the literature according to the inclusion and exclusion criteria, excluded irrelevant literature, and read the full text of the literature for possible inclusion criteria. In case of disagreement, it was resolved by discussion or consultation with the third researcher (Zhao). The included literature was finally extracted for information according to a pre-designed information extraction form, which included (1) essential characteristics of the literature, such as title, author, and year; (2) study quality, such as randomization method, allocation concealment method, blinding method, attrition, and/or dropout status; (3) subject baseline; (4) interventions; (5) outcome indicators, such as TAC, GPX, MDA, SOD.

### 2.5. Quality Assessment

The risk of RCT bias was evaluated using version 2 of the Cochrane tool for assessing the risk of bias in randomized trials (RoB2) [21], The overall assessment was based on 7 criteria, classified as high, moderate, or low risk of bias. We assess the risk of bias. Each question was considered and classified as “low”, “high”, or “unclear” independently by 2 authors (Chen and Chen). If there was disagreement, it was referred to a third person (Zhao) for review, and then the results were discussed by the study group to determine results.

### 2.6. Statistical Analysis

Literature quality assessment was produced using the ROB-2 tool (https://www.riskofbias.info/, Last clicked the link on 27 June 2022), and a network meta-analysis of the results of the different interventions was performed using STATA 17.0 (produced by StataCorp, College Station, TX, USA. https://www.stata.com/, accessed on 27 June 2022). Studies that reported standard error of the mean (SE), confidence interval (CI), or quartile were converted to standard deviation (SD) for analysis. The net effect size was calculated as the difference between the measurements from baseline to the end of the intervention, i.e., Mean_change_ = Mean_post_ − Mean_pre_ and SD_change_ = SQRT [(SD_pre_^2^ + Sd_pos_^2^) − (2 × Corr × SD_pre_ × SD_post_)], where the correlation coefficient was set to 0.5 according to the Cochrane Collaboration Handbook guidelines. A random-effects model (REM) was used to conduct a paired meta-analysis of the included randomized controlled trials. Then, Stata was used to analyze the effects of direct and indirect interventions compared to each other. For all possible pairwise comparisons, standardized mean difference (SMD) with a 95% confidence interval (CI) was estimated using the multivariate meta-analysis approach in which the different comparisons in studies are treated as different outcomes accounting for the correlation introduced by multi-arm trials. Again, the interventions were ranked by outcome according to the surface probability of surface under the cumulative ranking (SUCRA) [22], ranging from 0 to 1, for 5000 iterations. High probabilities in SUCRA signified favorable treatments. Larger SUCRA values indicated that the intervention was more effective. MDA was ranked using min values to calculate the ranking as a positive indicator of decreasing levels. In contrast, TAC, GPX, and SOD were classified using max values to calculate the order as a positive indicator of increasing levels. In addition, a node-splitting analysis was performed to estimate inconsistencies by comparing the differences between direct and indirect evidence [23]. When *p* > 0.05, the consistency model was used; otherwise, the inconsistency model was selected. In case of inconsistency, inconsistent data were excluded and calculated again until the closed-loop consistency was good [24].

## 3. Results

### 3.1. Search and Screening

A total of 323 relevant papers were retrieved for this study, and 15 articles were finally included according to the inclusion and exclusion criteria. The screening process is shown in Figure 1.

### 3.2. Included Study Characteristics

A total of 852 subjects were involved in the 15 included studies, including 2 studies comparing WQX involving 60 individuals, 3 studies comparing BDJ involving 74 individuals, 7 studies comparing TJQ involving 199 individuals, 2 studies comparing YJJ involving 72 individuals, 1 study comparing Mawangdui Guiding (MWD) involving 40 individuals, and 15 studies comparing no exercise intervention involving 407 individuals (Table 1).

### 3.3. Analysis of Inconsistency and Detection of Publication Bias

The risk of bias plots and summaries are shown in the figures (Figure 2 and Figure 3). (1) Selection bias: Three studies used a computerized table of random numbers, and two studies did not mention randomization. Four studies indicated allocation concealment. (2) Implementation bias: Two mentioned blinding participants and staff, and two explained the risks to participants. (3) Measurement bias: Four studies stated independent testing; the rest stated this. (4) Missed visit bias: Nine studies did not have missed visits. (5) Reporting bias: There was no selective reporting in all studies. (6) Other bias: No other bias was mentioned in all studies.

### 3.4. Results of Network Meta-Analysis

With 6 interventions (including NEI) in 15 studies, 15 pairs of intervention comparisons formed the network diagram. Five of the comparisons were direct comparisons, and 10 were indirect comparisons. A line between two points in the graph indicates evidence of a direct comparison between two interventions, and no line indicates evidence of indirect comparison. Among them, about the GPX indicator: There is 1 comparison between WQX and NEI, 5 direct comparisons between TJQ and NEI, 1 direct comparison between YJJ and NEI, and 1 direct comparison between MWD and NEI. There are indirect comparisons between WQX, TJQ, YJJ, and MWD. About the MDA indicator: There were 2 comparisons between WQX and NEI, 3 direct comparisons between BDJ and NEI, 2 direct comparisons between TJQ and NEI, 2 direct comparisons between YJJ and NEI, and 1 direct comparison between MWD and NEI indirect comparisons were made between WQX, BDJ, TJQ, YJJ, and MWD. About the SOD indicator: There were 2 comparisons between WQX and NEI, 3 direct comparisons between BDJ and NEI, 7 direct comparisons between TJQ and NEI, 2 direct comparisons between YJJ and NEI, and 1 direct comparison between MWD and NEI. Indirect comparisons were made between WQX, BDJ, TJQ, YJJ, and MWD (Figure 4). Only the TJQ study performed TAC was reported, and network comparisons of TAC indicators could not be completed.

#### 3.4.1. GPX

GPX indicators were reported by 462 subjects in 8 out of 15 investigations, with no GPX indicators recorded in the BDJ studies. Pairwise and network meta-analysis data for 8 RCTs are shown in Table 2. The included interventions are shown for two-way comparison in Table 2, and the comparisons are presented with SMD and 95% confidence intervals.

The results showed that the intervention effects of WQX (SMD = 2.79 (1.75, 3.83)), TJQ (SMD = 0.47 (0.23, 0.70)), YJJ (SMD = 1.78 (1.18, 2.37)), and MWD (SMD = 1.89 (1.36, 2.43)) were significantly better than those of NEI. In addition, WQX (SMD = 2.32 (1.26, 3.39)), YJJ (SMD = 1.31 (0.67, 1.95)), and MWD (SMD = 1.43 (0.84, 2.01)) were significantly better than TJQ (Table 2).

#### 3.4.2. MDA

MDA indices were reported in 10 investigations with a total of 576 participants. The results showed that the interventions of WQX (SMD = 1.68 (0.94, 2.42)) and YJJ (SMD = 0.99 (0.28, 1.69)) were significantly better than NEI. In addition, WQX was considerably better than TJQ (SMD = 1.47 (0.26, 2.67)) (Table 2).

#### 3.4.3. SOD

SOD indices were reported in 15 investigations with a total of 852 participants. The results showed that the intervention effects of WQX (SMD = 1.21 (0.55, 1.88)) and YJJ (SMD = 0.63 (0.16, 1.10)) were significantly better than those of NEI. In addition, the intervention effects of WQX were considerably better than those of YJJ (SMD = 1.52 (0.80, 2.24)) and MWD (SMD = 0.89 (0.03, 1.75)), and MWD intervention was significantly better than YJJ (SMD = 1.52 (0.80, 2.24)) (Table 2).

### 3.5. Ranking Probability

#### 3.5.1. GPX

The best intervention ranking can be understood in Figure 5. After 5000 iterations, the SCURA value shows the probability of each intervention being the best choice. The larger the SCURA, the higher the probability of being the best intervention. SUCRA of WQX: 0.97, MWD: 0.67, YJJ: 0.61, TJQ: 0.25, NEI: 0 (Figure 5). WQX is shown as the best way to intervene.

#### 3.5.2. MDA

SUCRA of WQX: 0.94, BDJ: 0.68, MWD: 0.51, YJJ: 0.42, TJQ: 0.40, NEI: 0.05 (Figure 5). WQX is shown as the best way to intervene.

#### 3.5.3. SOD

SUCRA of WQX: 0.91, MWD: 0.69, TJQ: 0.51, YJJ: 0.43, BDJ: 0.39, NEI: 0.05 (Figure 5). WQX is shown as the best way to intervene.

## 4. Discussion

After analyzing 15 RCTs examining traditional Chinese exercise, the present study found that all five traditional Chinese exercises were superior to NEI in improving oxidative stress in middle-aged and older adults. Traditional Chinese exercise can lead to decreased and increased pro-oxidative stress levels. Reports show that three out of five people worldwide suffer from non-communicable diseases caused by oxidative stress [36,37,38]. Antioxidant supplementation is thought to be a successful intervention. However, the clinical effectiveness of antioxidants has not been well established, and the question of which dose to use remains a problem for many scholars [8]. Meanwhile, the intake of antioxidants does not optimize the body’s oxidative and antioxidant balance without lifestyle changes [39]. On the other hand, regular physical activity has been shown to have sound effects in intervening oxidative stress [9] and is more effective with low to moderate-intensity aerobic exercise [10]. It is encouraging to note that traditional Chinese exercises are low to moderate-intensity aerobic sports [11,14,25,32,40]. In addition to that, traditional Chinese exercise has a compliance that conventional aerobic exercise does not have. Studies have shown that participants who performed traditional Chinese exercises had higher completion and greater adherence [41]. As a result, it is crucial to learn more about the effects of traditional Chinese exercise on oxidative stress. However, during the thousands of years of Chinese culture, traditional Chinese exercises have successively bred many gongfu movements with very different characteristics, gradually forming their unique exercise styles, rhythms, and intensities. Then, it is worthwhile for scholars to further explore whether there are differences in the effects of traditional Chinese exercise programs such as WQX, BDJ, TJQ, and YJJ on oxidative stress intervention in middle-aged and elderly people. Unfortunately, no relevant studies have been reported. Therefore, this is the first network meta-analysis comparing different traditional Chinese exercise interventions for oxidative stress in middle-aged and older adults.

The 15 studies included in this study showed that WQX was at the top of each intervention ranking, SUCRA of GPX: 0.97, MDA: 0.91, and SOD: 0.94. It is suggested that WQX has a better effect on improving oxidative stress in middle-aged and older adults. However, it is worth considering that because only indirect comparative evidence exists for the comparison of WQX with other intervention modalities, the conclusion of WQX as the best intervention modality needs to be explained with caution and further validated by subsequent studies. The mechanism by which WQX produces this effect may be related to its emphasis on breath control and regulation of mindfulness during exercise [42]. Improving the function of the respiratory system accelerates blood circulation [43], regulates the balance of the vegetative nerves [44], and controls the level of substance metabolism, thus accelerating lipid metabolism [45]. In contrast, psychomodulation enhances the excretion of lipid peroxidation products directly or indirectly through the regulation of hormones and improves the antioxidant capacity of the body [46]. Moreover, this study reported a statistically significant effect of WQX over TJQ, YJJ, and MWD. Specifically, the intervention effect of WQX on both GPX and MDA was significantly better than that of TJQ, with no significant difference in the remaining comparisons. It was hypothesized that the cause might be different exercise intensities. It was found that the heart rate was maintained at 120 beats/minute when participating in WQX [11]. When participating in TJQ, the heart rate change was around 100 beats/min [40]. When participating in YJJ and MWD, the heart rate change was 100 to 120 beats/min [14,24,31], which may be one of the factors contributing to the difference. Exercise intensity was discovered to be a significant determinant of oxidative stress levels [10], which is in line with the findings of this study. 

Among the SOD indicators, there was significance between WQX and MWD. It was found that WQX could directly increase the activity of catalase (CAT) and indirectly increase the rate of SOD production [46]. In contrast, there is no evidence yet in MWD studies indicating an additional boosting factor for SOD. Therefore, this may be the reason for the difference. In addition, this study also found significance between WQX, MWD, and YJJ in the SOD index, presumably because YJJ may require a longer intervention duration to obtain significant effects. A related study indicated that the 6-month YJJ intervention was not effective in enhancing SOD levels. In contrast, a significant increase in SOD levels was indicated in its follow-up report at 1 year [14]. The existence of trials with a 6-month intervention period in the YJJ studies included in this study may have contributed to the results of this study. It is suggested that the development of long-term healthy traditional Chinese exercise habits may be more helpful in improving oxidative stress levels and antioxidant capacity. Consistent with the conclusion that long-term aerobic exercise significantly improves oxidative stress status.

### Limitations of the Study

This study has innovative points but still has some shortcomings. (1) No other studies of traditional Chinese exercises were found: Six characters, Mulan Quan, Dawu; (2) studies have found that there are gender differences in the influence of exercise on oxidative stress. However, there is less gender division in the included literature, so it is not possible to analyze the image of the effect of exercise interventions by gender.

## 5. Conclusions

In typical traditional Chinese exercises, interventions employing WQX, TJQ, YJJ, and MWD can successfully improve oxidative stress and antioxidant capacity in middle-aged and older adults. WQX may be the best traditional Chinese exercise for reducing oxidative stress and increasing antioxidant capacity in middle-aged and older adults.

## Figures and Tables

**Figure 1 ijerph-19-08276-f001:**
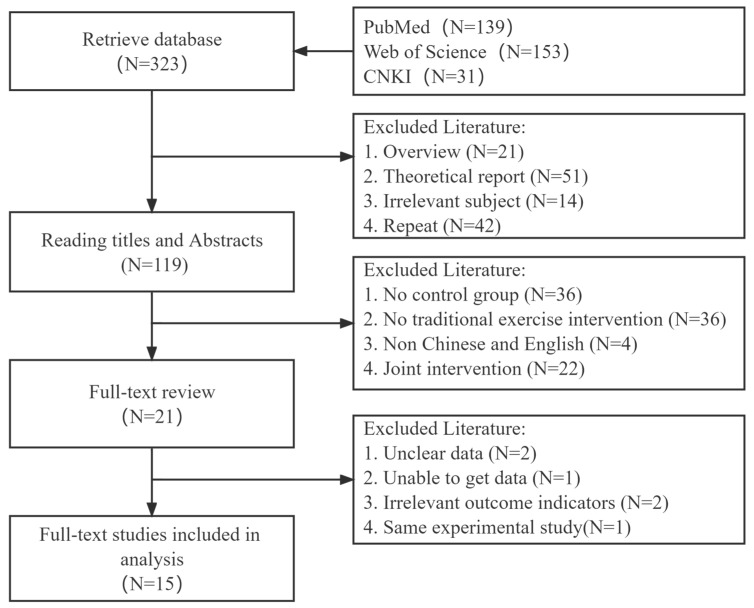
Literature screening flow chart.

**Figure 2 ijerph-19-08276-f002:**
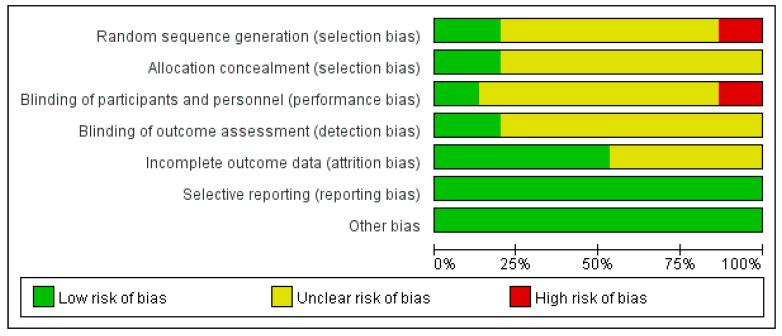
“Risk of bias” graph of included studies.

**Figure 3 ijerph-19-08276-f003:**
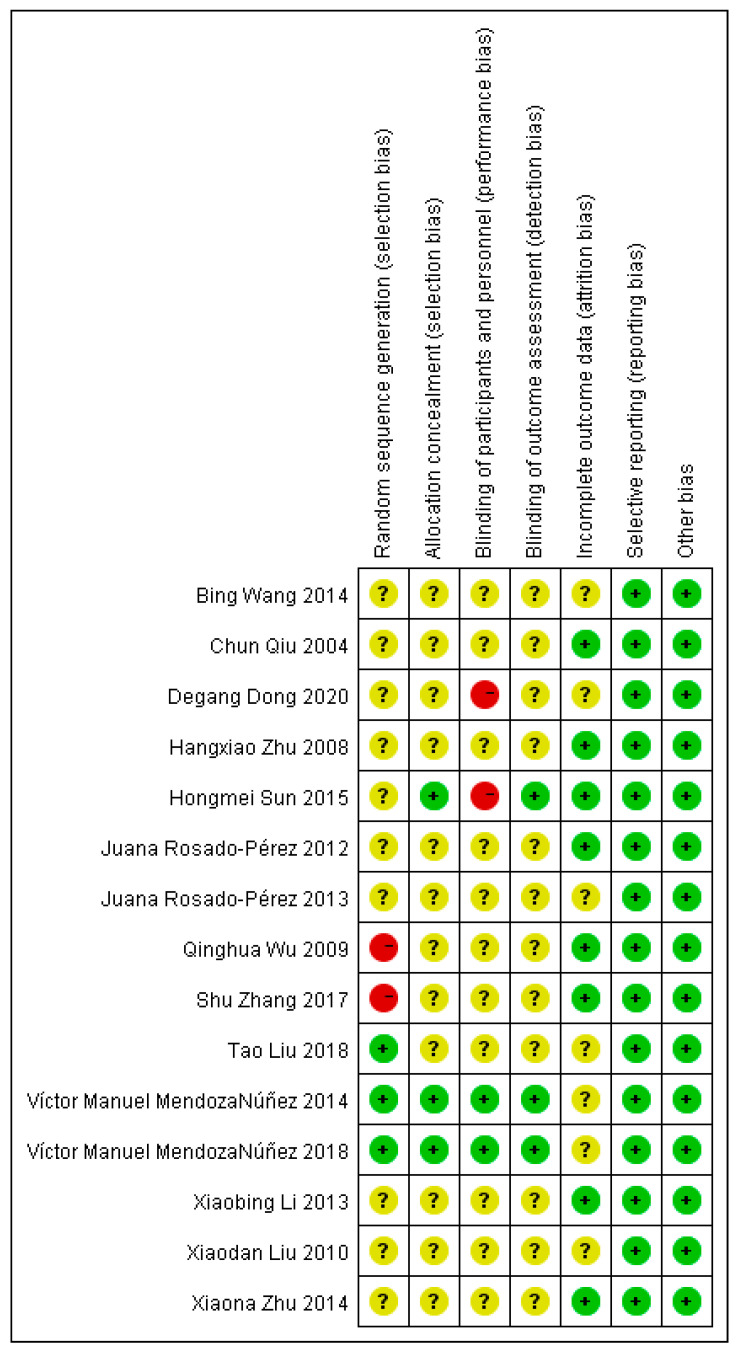
“Risk of bias” summary of included studies [11,12,13,14,25,26,27,28,29,30,31,32,33,34,35].

**Figure 4 ijerph-19-08276-f004:**
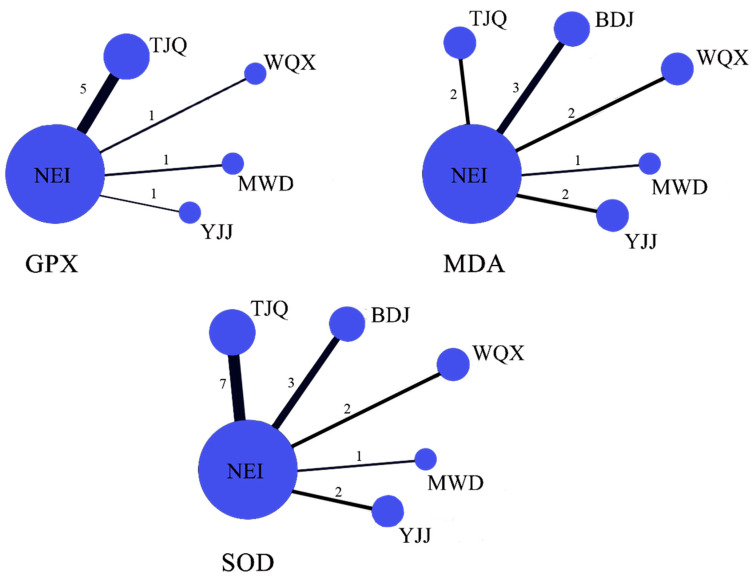
Network of treatment comparisons. (Notes: the width of the lines is proportional to the number of trials comparing every pair of treatments. The size of every circle is proportional to the sample size of the interventions; The numbers represent the number of studies that make direct comparisons between interventions. NEI, no exercise intervention; WQX, Wuqinxi; BDJ, Baduanjin; TJQ, Tai Ji Quan; YJJ, Yijinjing; MWD, Mawangdui Daoyin).

**Figure 5 ijerph-19-08276-f005:**
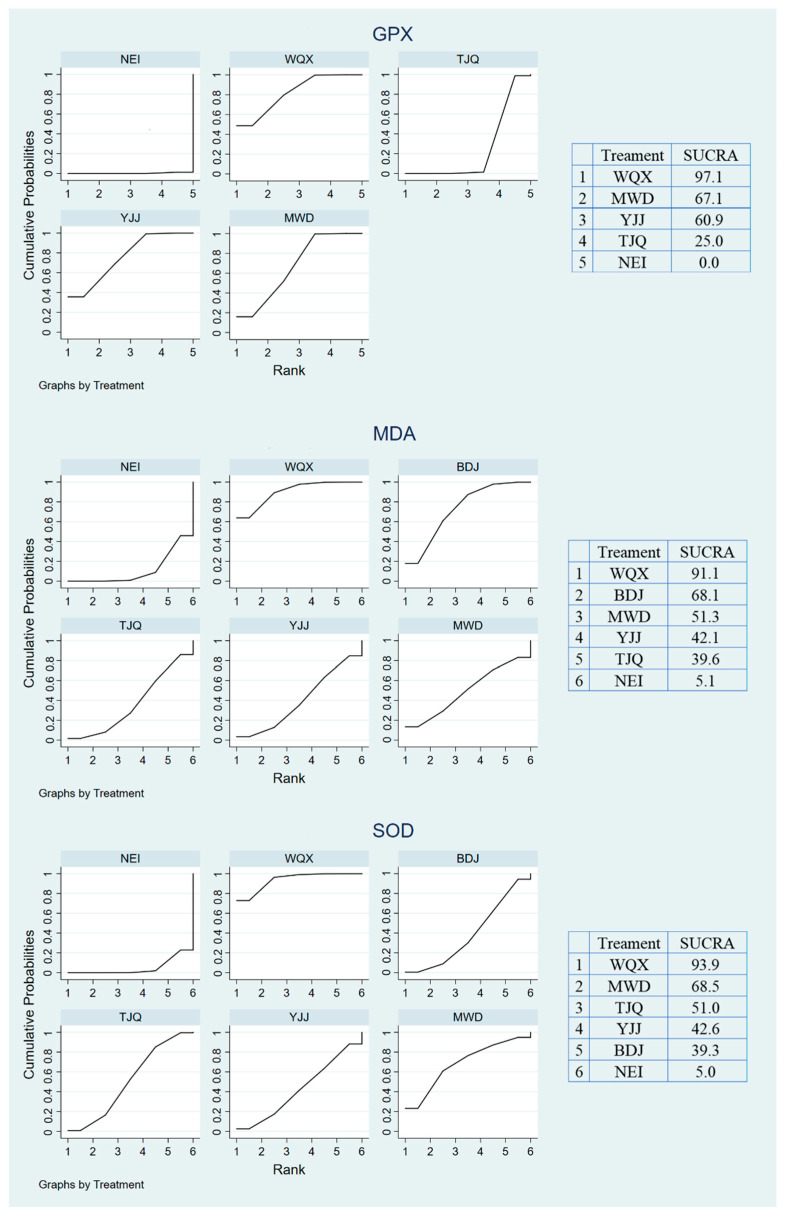
Results of the surface under the cumulative ranking and probability. (Notes: the table on the right shows the ranking of the best interventions. The area under the fold corresponds to the SUCRA value in the table on the right. The larger the SUCRA value, the more likely the intervention is to be the best choice; NEI, no exercise intervention; WQX, Wuqinxi; BDJ, Baduanjin; TJQ, Tai Ji Quan; YJJ, Yijinjing; MWD, Matangi Daoyin; GPX, glutathione peroxidase; MDA, malondialdehyde; SOD, superoxide dismutase).

**Table 1 ijerph-19-08276-t001:** Basic characteristics of included studies.

Reference	Interventions	Sex (Male/Female)	Age	Exercise Specifications (Single Time Duration, Frequency, Period)	Outcome
Bin Wang 2014 [25]	NEI	0/40	60.2 ± 4.1	1 h/d, 3 d/w, 20 w	GPX MDA SOD
	MWD	0/40	60.9 ± 5.1		
Chun Qiu 2004 [26]	NEI	17	58.7 ± 10.3	30 min/d, 6 d/w, 12 m	SOD
	TJQ	16	58.5 ± 10.6		
Degang Dong 2020 [12]	NEI	13/11	30–65	1 h/d, 5 d/w, 16 w	MDA SOD
	BDJ	13/10			
Hangxiao Zhu 2008 [27]	NEI	0/30	63.4 ± 1.8	45 min/d, 7 d/w, 16 w	MDA SOD
	WQX	0/45	62.8 ± 1.4		
Hongmei Sun 2015 [11]	NEI	15/0	40–50	40–50 min/d, 5 d/w, 6 m	GPX MDA SOD
	WQX	15/0			
Juana Rosado-Pérez 2012 [28]	NEI	23	N/A	50 min/d, 7 d/w, 6 m	TAC GPX SOD
	TJQ	32			
Juana Rosado-Pérez 2013 [29]	NEI	23	66.4 ± 4	1 h/d, 7 d/w, 6 m	TAC GPX SOD
	TJQ	31	66.7 ± 3.8		
Qinghua Wu 2009 [30]	NEI	7/37	62.8 ± 6.24	1 h/d, 5 d/w, 12 m	MDA SOD
	YJJ	8/32	62.85 ± 5.41		
Shu Zhang 2017 [31]	NEI	0/18	66.97 ± 3.21	45–55 min/d, 5 d/w, 4 m	GPX MDA SOD
	TJQ	0/18	67.77 ± 4.13		
Tao Liu 2018 [32]	NEI	17/13	71.6 ± 5.29	1 h/d, 6 d/w, 6 m	MDA SOD
	BDJ	21/9	71.23 ± 5.53		
Víctor Manuel MendozaNúñez 2014 [33]	NEI	25	60–74	1 h/d, 5 d/w, 6 m	TAC SOD
	TJQ	24			
Víctor Manuel MendozaNúñez 2018 [13]	NEI	37	68.2 ± 6.6	50 min/d, 5 d/w, 6 m	TAC GPX SOD
	TJQ	48	67.4 ± 4.7		
Xiaobin Li 2013 [34]	NEI	36/24	57.3 ± 10.3	45 min/d, 7 d/w, 8 w	GPX MDA SOD
	TJQ				
Xiaodan Liu 2010 [14]	NEI	0/30	65.7 ± 3.1	40–50 min/d, 6 d/w, 6 m	GPX MDA SOD
	YJJ	0/32			
Xiaona Zhu 2014 [35]	NEI	0/21	55.8 ± 4.67	1 h/d, 5 d/w, 6 m	MDA SOD
	BDJ	0/21	53.9 ± 4.05		

Notes: NEI, no exercise intervention; WQX, Wuqinxi; BDJ, Baduanjin; TJQ, Tai Ji Quan; YJJ, Yijinjing; MWD, Mawangdui Daoyin; min, minute; h, hour; d, day; w, week; m, month; TAC, total antioxidant capacity; GPX, glutathione peroxidase; MDA, malondialdehyde; SOD, superoxide dismutase.

**Table 2 ijerph-19-08276-t002:** Results of network meta-analysis.

**GPX**					
WQX					
**2.32 (1.26, 3.39)**	TJQ				
1.01 (−0.19, 2.21)	**−** **1.31 (−1.95, −0.67)**	YJJ			
0.89 (−0.27, 2.06)	**−1.43 (−2.01, −0.84)**	0.12 (−0.68, 0.92)	MWD		
**2.79 (1.75, 3.83)**	**0.47 (0.23, 0.70)**	**1.78 (1.18, 2.37)**	**1.89 (1.36, 2.43)**	NEI	
**MDA**					
WQX					
−0.76 (−2.30, 0.77)	BDJ				
**−** **1.47 (−2.67, −0.26)**	−0.70 (−2.36, 0.95)	TJQ			
**−** **1.35 (−2.08, −0.62)**	−0.59 (−1.94, 0.76)	−0.12 (−1.07, 0.84)	YJJ		
−1.00 (−2.08, 0.08)	−0.24 (−1.80, 1.33)	−0.47 (−1.71, 0.77)	−0.35 (−1.15, 0.44)	MWD	
−0.36 (−1.38, 0.66)	0.40 (−1.12, 1.92)	−1.10 (−2.29, 0.09)	**−** **0.99 (−1.69, −0.28)**	−0.63 (−1.70, 0.43)	NEI
**SOD**					
WQX					
0.51 (−0.89, 1.92)	BDJ				
1.01 (−0.11, 2.12)	0.49 (−0.98, 1.97)	TJQ			
**1.52 (0.80, 2.24)**	1.00 (−0.20, 2.21)	0.51 (−0.35, 1.36)	YJJ		
**0.89 (0.03, 1.75)**	0.37 (−0.92, 1.66)	−0.12 (−1.10, 0.85)	**−0.63 (−1.10, −0.16)**	MWD	
**1.05 (0.10, 2.01)**	0.54 (−0.82, 1.89)	0.04 (−1.02, 1.10)	−0.47 (−1.09, 0.16)	0.17 (−0.62, 0.95)	NEI

Notes: the table shows the included interventions for two-by-two comparison, and the comparison results are presented with SMD and 95% confidence intervals. Bold indicates a significant correlation between the two interventions; NEI, no exercise intervention; WQX, Wuqinxi; BDJ, Baduanjin; TJQ, Tai Ji Quan; YJJ, Yijinjing; MWD, Mawangdui Daoyin; GPX, glutathione peroxidase; MDA, malondialdehyde; SOD, superoxide dismutase.

## Data Availability

Not applicable.

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
