# Peer review of "Effects of Traditional Chinese Exercise on Oxidative Stress in Middle-Aged and Older Adults: A Network Meta-Analysis"

_ijerph, 2022, doi:10.3390/ijerph19148276_

Round 1

Reviewer 1 Report

The manuscript by Chen et al. renders strength in that the authors evaluate a research question that has not previous been evaluated. The manuscript is relevant for the field but may benefit from restructuring, clarification, and streamlining as it was difficult to follow the authors’ thought process, results, and methods. I greatly appreciate the authors taking this journey as systematic reviews/metanalyses are a major undertaking that allows us all a much better picture of the totality of evidence. Please consider my comments below as feedback for making the manuscript more easily interpreted by the reader:

Specific Comments:

·      Line 17: There are a few grammatical errors in this sentence and double use of “looked” and assumed you looked for all RCT that fit your search criteria not “a randomized controlled trial”?

·      Line 21: It may read easier to use a colon after “six interventions” and then list all.

·      Was the author’s study protocol registered in a database like PROSPERO as suggested by PRISMA guidelines? Include in methods and possibly abstract if so.

·      You may want to include the total number of papers reviewed, a bit more description of your population of interest (e.g. age range, healthy, etc.) and the specific oxidative stress outcomes in the abstract for the reader.

·      It is not clear what data you are summarizing in the abstract. SMD (and assuming) confidence interval should be defined and described in the method section and defined in the abstract.

·      Line 41: You may want to build your argument as it is not clear to the reader what the difficult situation is when you note “Faced with such a difficult situation…”.

·      Line 42: Suggest removing the dual use of “intervention”.

·      Line 43: You may want to expand and include references here.

·      Line 44: What is meant by exercise is a combination of good medicine and physical therapy? Also why are young people called out here given the population evaluated? Can you expand further?

·      Line 49: Is this in reference to the reduction of oxidative stress markers? It would help the reader to expand and provide a reference or perhaps this should be the first sentence of the next paragraph?

·      Given the international audience it might benefit the reader to have a short description of each of the workouts.

·      The introduction would benefit from briefly describing the mechanistic role for oxidative stress and chronic disease and any evidence on reduction and outcomes.

·      Line 51: It might be more beneficial to state a hypothesis rather than a question and move it to the last paragraph of the Introduction.

·      It is unclear what is meant by “CNKI was limited to core journals, CSSCI”. Is this well known for your readers?

·      The databases in the method section do not match the abstract.

·      Line 81: Were age ranges defined? What is your rationale for including all of these groups? Middle-aged is noted twice? Also, what is meant by etc. in this statement? Were other populations included?

·      Line 72: How were these exercises chosen?

·      Line 82: Was a minimum intervention duration defined?

·      Line 89: What is meant by “the same pilot study”?

·      Line 98: By “missed visits”, do you mean compliance/adherence?

·      Table 1: please define all abbreviations in the footnotes

·      Figure 1: Suggest “full-text review” or something similar instead of “read full text again”.

·      Figure 1: Suggest “Full-text studies included in analysis” or something similar instead of “Finally included in the literature”.

·      Figures 4 and 5 should have enough description in the legend to stand alone including abbreviations. Please include descriptions in the methods section as well. In my version of the image for figure 4 one of the exercises, likely MWD, is missing text. For example, “tolerability”.

·      Table 2 should be able to stand alone and needs descriptions of the values, brief summary of the analysis, and abbreviations in the footnotes section.

·      The results section would also benefit from noting what comparison were not significant. This may be in Table 2 but it is unclear based on the lack of description.

·      Line 169: What is meant by “best treatments” in the results section.

·      The discussion may benefit from inclusion of any other relevant systematic reviews on other exercise/workout/activity regimens and oxidative stress markers.

·      Line 223: Do you mean probabilities of reducing the oxidative stress markers….? Please clarify here and in the abstract.

·      Line 252-253: It was unclear that this analysis was complete. Please clarify in the methods and results sections.

·      Line 258: I would note, “Of the exercises evaluated, WQX was the best for reducing….”

Author Response

Thank you very much for your review and suggestions for this study, and here are the changes regarding your suggestions.

Point 1: Line 17: There are a few grammatical errors in this sentence and double use of “looked” and assumed you looked for all RCT that fit your search criteria not “a randomized controlled trial”?

Response 1: Grammatical errors have been corrected and language expressions have been optimized.

Point 2: Line 21: It may read easier to use a colon after “six interventions” and then list all.

Response 2: Added ":".

Point 3: Was the author’s study protocol registered in a database like PROSPERO as suggested by PRISMA guidelines? Include in methods and possibly abstract if so.

Response 3: The registration number has been completed and is located in the summary and methods. PROSPERO identifier: CRD42022332724. 

Point 4: You may want to include the total number of papers reviewed, a bit more description of your population of interest (e.g. age range, healthy, etc.) and the specific oxidative stress outcomes in the abstract for the reader.

Response 4: This paper has been made to include the total number of experiments with the total number of subjects. Regarding the age classification, there is no uniform standard for the age classification of middle-aged people, and multiple criteria exist. To avoid controversy, the age of 30 years is used as a cut-off for dividing the middle-aged population.

Point 5: It is not clear what data you are summarizing in the abstract. SMD (and assuming) confidence interval should be defined and described in the method section and defined in the abstract.

Response 5: The summary data are the concentration changes of the measured indicators.

Point 6: Line 41: You may want to build your argument as it is not clear to the reader what the difficult situation is when you note “Faced with such a difficult situation…”.

Response 6: has been replaced with a clearer representation of(Line 46).

Point 7: Line 42: Suggest removing the dual use of “intervention”.

Response 7: "Intervention" has been removed as per your suggestion.

Point 8: Line 43: You may want to expand and include references here. Line 44: What is meant by exercise is a combination of good medicine and physical therapy? Also why are young people called out here given the population evaluated? Can you expand further?

Response 8: Expanded and refined to strengthen the evidence that exercise is an excellent non-pharmacological treatment.

Point 9: Line 49: Is this in reference to the reduction of oxidative stress markers? It would help the reader to expand and provide a reference or perhaps this should be the first sentence of the next paragraph?

Response 9: Optimized the sentence and moved to the next paragraph(Line 57).

Point 10: Given the international audience it might benefit the reader to have a short description of each of the workouts.

Response 10: Appropriate characterization of different traditional Chinese sports has been done(Line 62).

Point 11: The introduction would benefit from briefly describing the mechanistic role for oxidative stress and chronic disease and any evidence on reduction and outcomes.

Response 11: The mechanisms of oxidative stress generation have been described, along with a description of the population selected for the study(Line 68).

Point 12: Line 51: It might be more beneficial to state a hypothesis rather than a question and move it to the last paragraph of the Introduction.

Response 12: Assumptions have been redescribed and placed in the last paragraph(Line 76).

Point 13: It is unclear what is meant by “CNKI was limited to core journals, CSSCI”. Is this well known for your readers?

Response 13: Because the quality of experiments in non-core journals is poor in the CNKI database, only core journals and CSSCI were searched in CNKI to assure the quality of publications in this study.

Regarding this issue, the authors also searched for non-core journals during the revision period and did not retrieve experiments that could be included in non-core journals; therefore, the description of this section was removed to reduce ambiguity.

Point 14: The databases in the method section do not match the abstract.

Response 14: Correction according to your suggestion.

Point 15: Line 81: Were age ranges defined? What is your rationale for including all of these groups? Middle-aged is noted twice? Also, what is meant by etc. in this statement? Were other populations included?

Response 15: Defined. “etc.” was described to prevent search omissions and has been corrected.

Point 16: Line 72: How were these exercises chosen?

Response 16: It is searched according to the Chinese traditional sports promulgated by the General Administration of Sports of China. Due to the Chinese and English translation problems, which resulted in the search with direct translation between Pinyin and English, it was necessary to confirm a large number of search terms with different names for the same sport.

Point 17: Line 82: Was a minimum intervention duration defined?

Response 17: Has been defined as a long-cycle intervention of more than 4 weeks.

Point 18: Line 89: What is meant by “the same pilot study”?

Response 18: The sentence has been re-described

Point 19: Line 98: By “missed visits”, do you mean compliance/adherence?

Response 19: The sentence has been re-described.

Point 20: Table 1: please define all abbreviations in the footnotes

Response 20: Correction according to your suggestion.

Point 21: Figure 1: Suggest “full-text review” or something similar instead of “read full text again”.

Response 21: Correction according to your suggestion.

Point 22: Figure 1: Suggest “Full-text studies included in analysis” or something similar instead of “Finally included in the literature”.

Response 22: Correction according to your suggestion.

Point 23: Figures 4 and 5 should have enough description in the legend to stand alone including abbreviations. Please include descriptions in the methods section as well. In my version of the image for figure 4 one of the exercises, likely MWD, is missing text. For example, “tolerability”.

Response 23: Correction according to your suggestion.

Point 24: Table 2 should be able to stand alone and needs descriptions of the values, brief summary of the analysis, and abbreviations in the footnotes section. The results section would also benefit from noting what comparison were not significant. This may be in Table 2 but it is unclear based on the lack of description.

Response 24: Table 2 with Figure 5, the description section is located at (Line 189). New content is added to explain.

Point 25: Line 169: What is meant by “best treatments” in the results section.

Response 25: The "best treatments" are "SUCRA". Correction according to your suggestion.

Point 26: The discussion may benefit from inclusion of any other relevant systematic reviews on other exercise/workout/activity regimens and oxidative stress markers.

Response 26: The arguments of related studies have been added and corroborated with the results of this study. Also, the objectives of the study are presented at the beginning of the discussion.

Point 27: Line 223: Do you mean probabilities of reducing the oxidative stress markers….? Please clarify here and in the abstract.

Response 27: It is a re-depiction of "SUCRA" to argue that the WQX is ranked first among the included interventions.

Point 28: Line 252-253: It was unclear that this analysis was complete. Please clarify in the methods and results sections.

Response 28: The section has been redescribed.

Point 29: Line 258: I would note, “Of the exercises evaluated, WQX was the best for reducing….”

Response 29: Correction according to your suggestion.

I have made changes in the original text based on your suggestions for revision. If you cannot understand my response, you can ask the editor to obtain my revision.

Thank you again for your suggestion.

Reviewer 2 Report

ABSTRACT

17 look twice..

Define the PICO in the methods, the outcome is missing how was the reduction of oxidative stress measured?

The systematic review resulted in the inclusion of 15 articles not the meta-analysis.

I would report the sucra of the interventions for greater clarity

INTRODUCTION

I would suggest broadening the introduction, concluding the section with the study objective not the outcome measure (move them in method section)

83 Did the controls have to include only other traditional Chinese medicine interventions? So there is no real reference "control"? Better to say .. any intervention of traditional Chinese medicine or no intervention

100 ROB-2

119 I suggest adding the description of net league table and references “Thus in a ranking table, Treatments were ranked from best to worst along the leading diagonal, below the leading diagonal were estimates from network meta-analyses.”
Ref: https://pubmed.ncbi.nlm.nih.gov/32668206/ 

184 Before introducing a figure, anticipate it with a few lines of manuscript to represent it and propose it to the reader with more reading tools. It is not easy for those who are not used to this methodology to understand the ranking tables or the sucra. In addition to the title it is necessary to explain everything that is in a figure, including abbreviations.

The discussion should begin with a paraphrase of the study objective

Author Response

Thank you very much for your review and suggestions for this study, and here are the changes regarding your suggestions.

Point 1: 17 look twice.

Response 1: Correction according to your suggestion.

Point 2: Define the PICO in the methods, the outcome is missing how was the reduction of oxidative stress measured?

Response 2: Correction according to your suggestion.

Point 3: I would report the sucra of the interventions for greater clarity

Response 3: Correction according to your suggestion.

Point 4: I would suggest broadening the introduction, concluding the section with the study objective not the outcome measure (move them in method section)

Response 4: ①Added the argument that exercise is a good non-pharmacological intervention.② Added a brief description of different traditional Chinese exercises.③ Added the mechanism of oxidative stress generation and the logic of the selection of study subjects. ④The study objectives were redescribed. ⑤Metrics have been moved to the methods section.

Point 5: 83 Did the controls have to include only other traditional Chinese medicine interventions? So there is no real reference "control"? Better to say .. any intervention of traditional Chinese medicine or no intervention

Response 5: The text was redescribed.——(3) Control: There was at least one control group with the control intervention of no exercise intervention (NEI).

Point 6: 119 I suggest adding the description of net league table and references “Thus in a ranking table, Treatments were ranked from best to worst along the leading diagonal, below the leading diagonal were estimates from network meta-analyses.”

Ref: https://pubmed.ncbi.nlm.nih.gov/32668206/

Response 6: Figure 5 has been replaced according to the reference you provided.

Point 7: 184 Before introducing a figure, anticipate it with a few lines of manuscript to represent it and propose it to the reader with more reading tools. It is not easy for those who are not used to this methodology to understand the ranking tables or the sucra. In addition to the title it is necessary to explain everything that is in a figure, including abbreviations.

Response 7: A description of Table 2 and Figure 5 has been added.

Point 8: The discussion should begin with a paraphrase of the study objective

Response 8: The arguments of related studies have been added and corroborated with the results of this study. Also, the objectives of the study are presented at the beginning of the discussion.

I have made changes in the original text based on your suggestions for revision. If you cannot understand my response, you can ask the editor to obtain my revision.

Thank you again for your advice!

Round 2

Reviewer 1 Report

Your careful attention to revision is much appreciated, thank you. The manuscript would benefit from grammatical editing in addition to the updates based on the comments below.

Specific Comments:

·      Line 16-18: Requires further grammatical revision.

·      Line 20 and 21: Why are certain markers capitalized?

·      Line 23: This sentence is incomplete.

·      Line 28-29: Do you mean the noted exercises were superior in reducing MDA relative to NEI? The question applies to the next sentence as well.

·      Line 30: It would help the reader to introduce SUCRA earlier.

·      Line 31: Do you mean the WQX intervention had a greater effect on GPX…?

·      Line 34: Suggest revision to include “of the exercises evaluated”

·      Line 84: Can you confirm? International prospective register of systematic reviews?

·      Line 110: At least one of the following oxidative stress markers…?

·      Line 151: What is meant by “safer”?

·      Table 1 footnotes: I would be consistent with either sex (female/male) or gender (women/men). You may also want to explain the exercise specifications format (e.g., session, repetition, duration is that correct)?

·      Table 1: You could consider listing your outcomes instead of number for ease of the reader.

·      Line 203 and 206: Are these percentages or do values range from 1 to 100?

·      Line 202-205: Should this be in the methods section?

·      Tables and Figures: Please ensure your tables and figures can stand alone with the provided legends/footnotes. For example there should be a short statement on the analysis and what data is being presented and also clear descriptions on the content for example, the bolded values in Table 2.

Author Response

Response to Reviewer 1 Comments

Thank you again for your suggestions, and I have made changes to this study based on your suggestions.

Point 1: Line 16-18: Requires further grammatical revision.

Response 1: The problem with the syntax has been fixed.

Point 2: Line 20 and 21: Why are certain markers capitalized?

Response 2: The incorrect capitalization has been corrected.

Point 3: Line 23: This sentence is incomplete.

Response 3: Missing sentences have been added, and the description of SCURA has been added. "Standardized mean differences (SMD) and 95% confidence intervals (CI) were used to assess the correlation between each group of interventions, and Surface Under the Cumulative ranking (SUCRA) was used to rank the best interventions.”

Point 4: Line 28-29: Do you mean the noted exercises were superior in reducing MDA relative to NEI? The question applies to the next sentence as well.

Response 4: Additional sentences were added to make it easier to understand.

Point 5: Line 30: It would help the reader to introduce SUCRA earlier.

Response 5: Description has been added, in line 26.

Point 6: Line 31: Do you mean the WQX intervention had a greater effect on GPX…?

Response 6: Replaced sentences with easier-to-read ones.

Point 7: Line 34: Suggest revision to include “of the exercises evaluated”

Response 7: Added based on comments.

Point 8: Line 84: Can you confirm? International prospective register of systematic reviews?

Response 8: I confirm that the registration number is searchable.

Point 9: Line 110: At least one of the following oxidative stress markers…?

Response 9: The expression has been modified.

Point 10: Line 151: What is meant by “safer”?

Response 10: Ambiguous expressions have been removed.

Point 11: Table 1 footnotes: I would be consistent with either sex (female/male) or gender (women/men). You may also want to explain the exercise specifications format (e.g., session, repetition, duration is that correct)?

Response 11: Added and modified

Point 12: Table 1: You could consider listing your outcomes instead of number for ease of the reader.

Response 12: Modified according to your suggestions.

Point 13: Line 203 and 206: Are these percentages or do values range from 1 to 100?

Response 13: The range is 0-1, and the presence of percentage expressions or numeric expressions has been changed to numeric expressions.

Point 14: Line 202-205: Should this be in the methods section?

Response 14: The expression has been modified to specifically describe the contents of Figure 5.

Point 15: Tables and Figures: Please ensure your tables and figures can stand alone with the provided legends/footnotes. For example there should be a short statement on the analysis and what data is being presented and also clear descriptions on the content for example, the bolded values in Table 2.

Response 15: Annotations have been added to the chart, along with a chart description, in lines 193-206 and 219-229.

Reviewer 2 Report

Dear Authors, the manuscript has improved, there are still small concerns about the methodological illustration

L23 Truncated phrase. In this regard it is difficult to define the best treatment with MDS, because this is only the effect size of each intervention versus an overall control generated by the network meta-analysis. Or they are head-to-head comparisons between interventions but this distorts the concept of best treatments among all. it is better to define the ranking based on the SUCRA score

L127 Refer to the RoB-2 and I suggest you make a reference

L137 missing manufacturer, city and state

L148 I suggest adding the description of net league (ranking tables) and reference “Thus in a ranking table, Treatments were ranked from best to worst along the leading diagonal. Above the leading diagonal were estimates from pairwise meta-analyses, below the leading diagonal were estimates from network meta-analyses.” Reference: https://pubmed.ncbi.nlm.nih.gov/33221632/

Figure 4. Figures and tables stand on their own. So, I suggest describing them to make readers understand what lines and circles represent. Unfortunately for those who lead networks it seems trivial, for newbies it may be difficult to understand even the most trivial representations

Table 2. As above I suggest describing the nature of the table, if anything referring to what is suggested in line 148.

In figure 5 I suggest repeating the concept of SUCRA and the fact that the higher its value, the more likely intervention is the best possible option.

Author Response

Thank you again for your suggestions, and I have made changes to this study based on your suggestions.

Point 1: L23 Truncated phrase. In this regard it is difficult to define the best treatment with MDS, because this is only the effect size of each intervention versus an overall control generated by the network meta-analysis. Or they are head-to-head comparisons between interventions but this distorts the concept of best treatments among all. it is better to define the ranking based on the SUCRA score

Response 1: The sentence was completed with the addition of the description of SUCRA.

Point 2: L127 Refer to the RoB-2 and I suggest you make a reference

Response 2: Added a reference to ROB-2, in line 131.

Point 3: L137 missing manufacturer, city and state

Response 3: Manufacturers and download sites have been added.

Point 4: L148 I suggest adding the description of net league (ranking tables) and reference “Thus in a ranking table, Treatments were ranked from best to worst along the leading diagonal. Above the leading diagonal were estimates from pairwise meta-analyses, below the leading diagonal were estimates from network meta-analyses.” Reference: https://pubmed.ncbi.nlm.nih.gov/33221632/

Response 4: Added corresponding comments and descriptions, in line 227.

Point 5: Figure 4. Figures and tables stand on their own. So, I suggest describing them to make readers understand what lines and circles represent. Unfortunately for those who lead networks it seems trivial, for newbies it may be difficult to understand even the most trivial representations

Response 5: Added corresponding comments and descriptions, in line 193.

Point 6: Table 2. As above I suggest describing the nature of the table, if anything referring to what is suggested in line 148.

Response 6: The corresponding description has been added, in lines 219-225.

Point 7: In figure 5 I suggest repeating the concept of SUCRA and the fact that the higher its value, the more likely intervention is the best possible option.

Response 7: Notes have been added to the legend.

Round 3

Reviewer 2 Report

The manuscript has improved significantly and I can suggest suitability to publication